# Prompt Reverse Learning: Enhancing Visual Language Models for Rare Image Recognition

## Abstract

Large visual language models like CLIP have demonstrated impressive performance on various downstream tasks involving common data, e.g., natural images, by leveraging prompt learning. However, these models often falter when applied to tasks involving rare data, e.g., medical images. We provide an experimental insight into this phenomenon: CLIP is insensitive to the class names of rare images. For instance, replacing the class name "medulloblastoma" (a type of brain tumor) with "dog" in prompts has minimal impact on performance, a phenomenon not observed with common images. This highlights the disparity in representation learning between common and rare data. To realign prompt learning with rare image recognition, we propose a novel prompt learning strategy, termed **p**rompt **r**everse **le**ar**n**ing (PeLen). Different from the existing methods that *adapt CLIP's representations to downstream tasks*, PeLen *adapts task-specific representations to CLIP's representations*. Built upon the insensitivity to the class names of rare images, PeLen designates common images and their class names to represent a specific class of rare images and class names, e.g., allowing the image and text of a dog to correspond to the image and text of medulloblastoma. Consequently, PeLen learns prompts to align the representations between the rare images and the visual and textual representations of common images. Our experiments on three types of rare images demonstrate the efficacy of PeLen for rare image recognition.

## 1 Introduction

Contrastive Language-Image Pre-training (CLIP) Radford et al. (2021) stands out as a significant milestone in large-scale language vision models, drawing considerable attention for its exceptional performance in various tasks. CLIP intricately integrates images and text, embedding them into a shared representation space through extensive training on large-scale datasets. In this representation space, the correlations between images and text are precisely captured and encoded, enabling CLIP to simultaneously comprehend the meaning of both images and text. This makes it possible to perform zero-shot inference, where CLIP leverages the rich knowledge learned during pre-training to make judgments by comparing and measuring the embeddings of images and text in the representation space. This multi-modal alignment capability empowers CLIP to excel not only in traditional visual tasks such as image classification Zhou et al. (2022a); Gao et al. (2024) and object detection Yao et al. (2022) but also in more complex image generation Ramesh et al. (2022) and language understanding Li et al. (2022a) tasks, demonstrating remarkable universality and adaptability.

Advanced research has extensively explored CLIP's alignment capabilities, particularly focusing on tuning the context (embeddings) of prompts, i.e., the text input of the text encoder. Specifically, since small changes in manual prompts can bring huge performance differences, it is particularly important to optimize the prompts. In this context, CoOp Zhou et al. (2022b) introduces learnable vectors into prompts, essentially learning a text embedding that better matches visual features extracted by the image encoder, thereby enhancing CLIP's performance and efficiency. Futher research such as MaPLe Khattak et al. (2023) further aligns the visual-language (V-L) modalities through multi-modal prompts to achieve deeper fusion and synergy. These emerging approaches contribute to strengthening CLIP's adaptability to various domains and tasks, sparking further exploration and innovation in prompt learning methods.

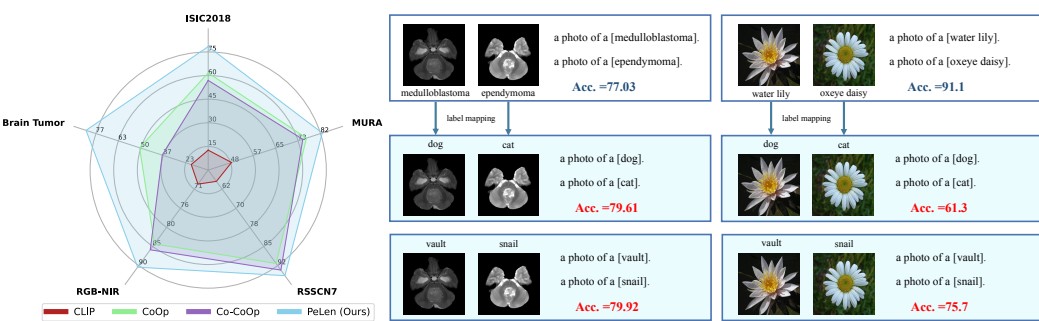

(a) Performance comparison  (b) Class names replacement in medical images and natural images

Figure 1: **Motivation illustration: CLIP is insensitive to the class names of rare images.** Figure 1a shows the performance comparison results of our method with CLIP Radford et al. (2021), CoOp Zhou et al. (2022b) and Co-CoOp Zhou et al. (2022a). The medical images in Figure 1b are pediatric brain tumor images while the natural images are from Flower12 Nilsback & Zisserman (2008). When replacing the tumor names with common image names, the performance change of prompt learning is minimal. However, replacing real labels with names unrelated to the images significantly diminishes the performance of prompt learning. This observation is corroborated by two sets of class name replacement experiments conducted on both medical and natural images.

However, prompt learning methods would face challenges in downstream tasks with rare images, such as medical, remote sensing, and infrared images. This is because these images rarely appear in the training dataset of CLIP, compared with common data, e.g., natural images. Accordingly, it could be difficult for CLIP to align rare images and the corresponding text in the shared representation space. Our experiments, as shown in Figure 1a, verify this point, where the zero-shot inference ability of CLIP slightly outperforms random guessing on rare images. Meanwhile, learning prompt Zhou et al. (2022b;a) for rare images brings limited performance gain, which is not as exciting as that on natural images Zhou et al. (2022b;a). While rare images occur less frequently in CLIP training data due to privacy or security concerns, adapting pre-trained vision-language models to tasks involving rare images remains crucial.

To re-enable prompt learning for rare image recognition, we perform an experimental exploration, motivated by the poor ability to align rare images and their captions. In the existing prompt learning methods, class names mainly serve as the class priors in the optimizing process to guide model learning. In the rare image scenario, the class name priors may hardly contribute to optimizing prompts due to the poor ability to align visual and textual features. Thus, replacing the class name of rare images with a dummy word would cause limited performance change, which distinguishes rare images from natural images. Fortunately, our experimental results provide a consistent conclusion with the point, as depicted in Figure 1b. Specifically, when optimizing context, we replace the class names of rare images in prompts with those of common images. For instance, we leverage "dog" as the class name for the MRI image of "medulloblastoma" (a type of brain tumor) to optimize prompts. Our results show that even dummy words can achieve performance comparable to the official class names. Namely, the performance of prompt learning is insensitive to the class names when processing rare images. By contrast, performance on natural images drops significantly when replacing "water lily" with "dog."

These experimental explorations motivate us to leverage the CLIP's insensitivity to class names of rare images to promote prompt learning. We propose **p**rompt **r**everse **le**ar**n**ing (PeLen) for prompt learning with rare images. The insight is that *PeLen adapts task-specific representations to CLIP's representations*, which is different from *existing methods adapting CLIP's representations to downstream tasks*. Specifically, PeLen first assigns a class of common images to each class of rare images to establish a mapping relationship, e.g., letting images and text of "dogs" correspond to images and text of "medulloblastoma". Subsequently, PeLen optimizes prompts to approach representations of rare images to those of common images. Meanwhile, PeLen predicts labels of rare images using the classification weight generated by prompts of common images. CLIP lacks prior knowledge of rare images, but has sufficient prior knowledge of common images. Consequently, PeLen promotes the

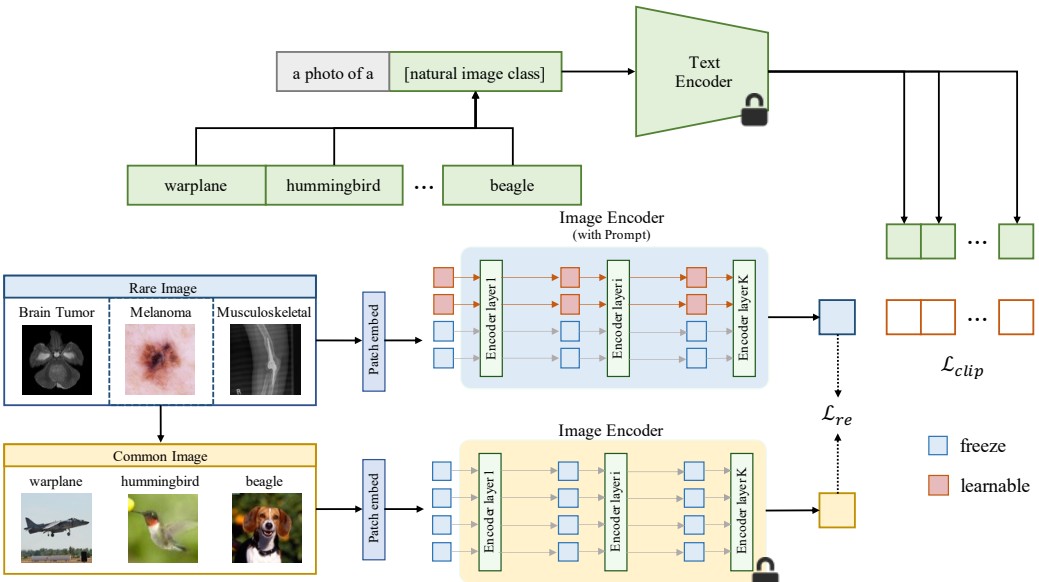

Figure 2: Overview of Prompt Reverse Learning (PeLen). PeLen specifies common images and their class names to represent rare images and class names of a specific class. PeLen uses deep visual prompts and only adjusts the visual branch of rare images but keeps the visual branch of common images frozen. Moreover, PeLen aligns the visual representation of rare images with the fixed visual and textual representation of common images through cross-entropy loss and reverse learning loss.

ability to align rare images by transforming rare image features into common ones for which CLIP has strong visual text matching capabilities.

To verify the proposed PeLen's effectiveness on rare images, we conduct experiments on three types of rare image tasks: medical, remote sensing, and infrared images. Moreover, we further validate PeLen on a real-world pediatric brain tumor dataset. Experimental results demonstrate that PeLen significantly outperforms the existing prompt learning methods, achieving state-of-the-art performance on rare image recognition.

Our main contributions can be summarized as follows,

- We point out the difficulties of the existing prompt learning methods on rare image recognition. Accordingly, we conduct experimental exploration to provide insight into this phenomenon: CLIP is insensitive to the class names of rare images, as depicted in Figure 1.

- Motivated by our experimental observations, we propose a novel approach, termed reverse learning (PeLen), to re-enable prompt learning for rare image recognition. Specifically, rather than adapting CLIP's representations to downstream tasks, PeLen adapts task-specific representations to CLIP's representations, see Eq. 10.

- Experiments on three types of rare images demonstrate that PeLen can significantly outperform the existing prompt learning methods. Moreover, we construct a real-world pediatric brain tumor dataset for evaluation, showing that PeLen can outperform baseline methods.

## 2 METHODOLOGY

We propose prompt reverse learning (PeLen) to enhance the generalization ability of the visual language model CLIP to downstream tasks of rare images. PeLen updates only a small number of parameters in the vision branch of CLIP, establishing connections between rare and common images. The overview of PeLen is shown in Figure 2. In this section, we first review the structure of CLIP and visual prompt and then give the details of the proposed PeLen.

## 2.1 PRELIMINARIES

### 2.1.1 CLIP STRUCTURE

CLIP consists of two branches: vision encoder branch and language encoder branch. Given an image $I$ and several sentences $\{S_i\}_{i=1}^K$ as input to the visual and language branches respectively, where $K$ represents the number of categories. We use "`a photo of a <category>`" as a prompt for text input and fill the class name in `<category>`. Each branch contains an encoder, namely image encoder $\mathcal{V}$ and text encoder $\mathcal{T}$.

**Image encoder.** Image encoder divides the image $I$ into $m$ fixed-size patches firstly. Then the patches are flattened and projected to fixed dimensions to obtain the image embedding $E_o \in \mathbb{R}^{m \times d_v}$. Concatenating with class embeddings and adding positional embeddings, image embedding is fed to the transformer, which has $L$ layers. Each transformer layer consists of a multi-head self-attention layer and a multi-layer perception block with layer normalization and residual connections. We denote $E_i$ as the image embedding output by the $i$-th transformer layer $\mathcal{V}_i$, which is then input to the $(i+1)$-th transformer layer $\mathcal{V}_{i+1}$ together with the class embedding $c_i$,

$$[c_i, E_i] = \mathcal{V}_i([c_{i-1}, E_{i-1}]) \quad i = 1, 2, \cdots, L. \tag{1}$$

The class embedding $c_L$ obtained by the last transformer layer is projected into a fixed-dimensional latent space to obtain the final image representation $\boldsymbol{x}$,

$$\boldsymbol{x} = \texttt{proj}(c_L). \tag{2}$$

**Text encoder.** After tokenizing the input sentences $\{S_i\}_{i=1}^K$, we obtain the corresponding token sequences, which are then mapped to their respective vector representations through an embedding layer. These vector representations are then fed into the transformer layer to generate the encoded representation of the sentence. Finally, the text encoder extracts the encoded representation corresponding to the last token of the sentence and maps it to the latent embedding space to obtain the final text representation $W = \left\{ \boldsymbol{w}_i \in \mathbb{R}^{d_m} \mid i \in \mathbb{N}, 1 \leq i \leq K \right\}$.

**Prediction and training.** After obtaining image representation $\boldsymbol{x}$ and text representations $W$ respectively, CLIP uses cosine similarity function to measure the similarity between image representation and text representations to get the predicted probability, as follows:

$$p(y \mid \boldsymbol{x}) = \frac{\exp\left(\text{sim}\left(\boldsymbol{x}, \boldsymbol{w}_y\right)/\tau\right)}{\sum_{i=1}^K \exp\left(\text{sim}\left(\boldsymbol{x}, \boldsymbol{w}_i\right)/\tau\right)}, \tag{3}$$

where $\tau$ and $\boldsymbol{w}_y$ represent temperature parameter and text representation of class $y$, respectively, and $\text{sim}(\cdot)$ stands for cosine similarity function. Combined with the cross-entropy loss, the loss function during training procedure is formulated by:

$$\mathcal{L}_{clip} = -\frac{1}{B} \sum_{i=1}^B \log \frac{\exp\left(\text{sim}\left(\boldsymbol{x}_i, \boldsymbol{w}_y\right)/\tau\right)}{\sum_{j=1}^K \exp\left(\text{sim}\left(\boldsymbol{x}_i, \boldsymbol{w}_j\right)/\tau\right)}, \tag{4}$$

where $B$ represents the size of a mini batch.

### 2.1.2 VISUAL PROMPT

VPT Jia et al. (2022) proposes a prompt learning method for large-scale visual Transformer models. CLIP's advanced prompt learning method MaPLe Khattak et al. (2023) also adopts the similar approach. Specifically, they insert a small number of learnable tokens into each transformer layer. We define the learnable tokens inserted in the $i$-th transformer layer $\mathcal{V}_i$ as $P_i = \left\{ \boldsymbol{p}_i^j \in \mathbb{R}^{d_v} \mid j \in \mathbb{N}, 1 \leq j \leq b \right\}$, $b$ is the length of the prompt. Similar to class embeddings, these learnable tokens are concatenated with image embeddings and then input to the $(i+1)$-th transformer layer $\mathcal{V}_{i+1}$. However, because of the correlation of feature processing to deeper transformer layers, prompts are no longer learned independently, as follows:

$$[c_i, E_i, \_\_] = \mathcal{V}_i([c_{i-1}, E_{i-1}, P_{i-1}]) \quad i = 1, \cdots, J, \tag{5}$$

$$[c_k, E_k, P_k] = \mathcal{V}_k([c_{k-1}, E_{k-1}, P_{k-1}]) \quad k = J+1, \cdots, L. \tag{6}$$

Here $J$ is the depth of the prompt. To obtain the final image representation $\boldsymbol{x}$, the class embeddings $c_k$ of the last layer are mapped to a fixed-dimensional latent space:

$$\boldsymbol{x} = \texttt{proj}(c_L). \tag{7}$$

## 2.2 PeLen: Prompt Reverse Learning

The existing prompt learning methods mainly adapt models to downstream tasks. In contrast, we propose a novel approach inspired by our experimental phenomenon, aiming to enhance the generalization of CLIP to rare data. Unlike previous works, which fine-tune prompts to adapt CLIP to downstream representation distributions, PeLen aligns downstream rare image representation distributions with fixed upstream image and text representation distributions, as shown in Figure 2.

### 2.2.1 Label and Image Mapping

We randomly assign common images and their class names to represent rare images of a specific class and their class names to establish the connection between rare data and common data. We choose ImageNet Russakovsky et al. (2015) as the common image dataset.

To be concrete, let $\mathcal{S}$ and $\mathcal{N}$ denote rare image dataset and common image dataset, respectively. The rare image dataset contains $K_1$ categories and a total of $N$ images, while the common image dataset contains $K_2$ categories. We randomly assign a common image class name to represent a rare image category, and define $\mathcal{F}(\cdot)$ as the label mapping function,

$$\mathcal{F}(y_k^S) = y_m^N \quad 1 \le k \le K_1, 1 \le m \le K_2, \tag{8}$$

where $y_k^S$ is the $k$-th class names of the rare image dataset and $y_m^N$ is the $m$-th class names of the common image dataset. This establishes the label-mapping relationship between common images and rare images. Then, PeLen randomly assigns a common image with label $y_k^S$ to the rare image with label $y_m^N$ to establish a mapping relationship between the images,

$$\mathcal{H}(I_{i,k}^S) = I_{j,m}^S \quad I_{i,k}^S \in \mathcal{S}, I_{j,m}^N \in \mathcal{N}, \tag{9}$$

where $I_{i,k}^S$ represents the image with $k$-th class name of the rare image dataset and $I_{j,m}^N$ represents the image with $m$-th class name of the common image dataset, $\mathcal{H}(\cdot)$ is image mapping function.

### 2.2.2 Reverse Learning

Since the training set of CLIP rarely contains rare images, previous prompt learning methods demonstrate impressive performance on downstream tasks with common images, but exhibit limitations when applied to rare image tasks. Our experiments reveal that CLIP shows insensitivity towards rare image classes, but demonstrating strong visual-textual matching capabilities for common images. Leveraging this attribute, we align the visual representations of rare images with the fixed visual and textual representations of common images respectively.

Given an image $I_{i,k}^S$ with $k$-th class name of the rare image dataset, through label and image mapping, we obtain its corresponding image $I_{j,m}^N$ with $m$-th class name of the common image dataset. Common image $I_{j,m}^N$ is input to the image encoder without prompt, which is the initial image encoder of CLIP, to obtain a fixed visual representation $\boldsymbol{x_j^N}$ of common images. Rare image $I_{i,k}^S$ is fed to the image encoder with prompt to obtain visual representation $\boldsymbol{x_i^S}$. We use cosine similarity to measure whether the rare image representation is aligned with its corresponding common image representation, and introduce a cosine similarity loss function during the training process,

$$\mathcal{L}_{re}(\theta) = -\frac{1}{B} \sum_{i=1}^{B} \left( 1 - \mathrm{sim}\left( \boldsymbol{x}_i^S(\theta), \boldsymbol{x}_j^N \right) \right), \tag{10}$$

where $\theta$ are learnable parameters. That establishes the correspondence between rare images of a specific class and common images at the visual level. Additionally, We replace rare image class name $k$ with the common image class name $m$ in language branch to align rare images and label-mapped common class name. With reference to Eq.4, the loss function can be expressed as:

$$\mathcal{L}_{clip}(\theta) = -\frac{1}{B} \sum_{i=1}^{B} \log \frac{\exp\left( \mathrm{sim}\left( \boldsymbol{x}_i^S(\theta), \boldsymbol{w}_y \right)/\tau \right)}{\sum_{i=j}^{K} \exp\left( \mathrm{sim}\left( \boldsymbol{x}_i^S(\theta), \boldsymbol{w}_j \right)/\tau \right)}, \tag{11}$$

Combining the two losses in Eq.10 and reverse Learning loss in Eq.11, PeLen optimizes visual prompt by minimizing loss:

$$\min_{\theta} \mathcal{L}(\theta) = \mathcal{L}_{re}(\theta) + \beta \times \mathcal{L}_{clip}(\theta), \tag{12}$$

where $\beta$ is a tunable hyperparameter.

## 3 RELATED WORK

### 3.1 VISION LANGUAGE MODELS

Recently, pre-trained visual language models (VLM) have made significant progress in joint learning of images and text. Models like CLIP Radford et al. (2021) and ALIGN Jia et al. (2021)utilize extensive image-text pairs to train multimodal networks, demonstrating impressive performance in various downstream tasks Gao et al. (2024); Zhang et al. (2021); Du et al. (2022); Wu et al. (2023) through contrastive learning. In the development of VLM, some of their variants further optimize the performance of the model. For instance, GLIP Li et al. (2022b) unifies object detection and phrase grounding tasks, incorporating image and text prompts simultaneously into the object detection network. CLIPSeg Lüddecke & Ecker (2022) and CRIS Wang et al. (2022) use CLIP for segmentation. Although the VLM model learns a wide range of general knowledge, it still faces challenges in the huge distribution differences between medical images and natural images. In this work, we propose a prompt reverse learning technique based on CLIP to effectively apply CLIP to rare image recognition.

### 3.2 PROMPT LEARNING

Prompt learning is initially introduced in NLP Jiang et al. (2020); Tsimpoukelli et al. (2021); Shin et al. (2020). By designing specific prompts, the model is guided to perform text understanding, generation or classification in the desired way without modifying the large-scale pre-trained model. Inspired by prompt learning in NLP, the VL model begin to replace manually designed context by automatically optimizing context. CoOp Zhou et al. (2022b) trains a set of learnable vectors for the language branch end-to-end to achieve better downstream performance and efficiency. Co-CoOp Zhou et al. (2022a) trains specific prompts for image instances, addressing the overfitting issue of CoOp on base classes and improving performance on new classes. Beyond language prompts, Bahng et al. (2022) learns perturbations on images to explore the effectiveness of visual prompts for large-scale Models. VPT Jia et al. (2022) introduces a small amount of image tokens as learnable prompts, enabling Vision Transformer Dosovitskiy et al. (2020) to adapt to downstream visual tasks. MaPLe Khattak et al. (2023) proposes multi-modal prompts, coupling visual and language branches to promote synergy and enhance the generalization ability of VLM. However, these prompt learning methods do not fully consider the problem of insufficient perception when rare images as the role of the downstream tasks. In this work, we leverage prompts to align the representations of rare images with the visual and textual representations of common images, thus taking full advantage of CLIP's powerful perceptual capabilities.

## 4 EXPERIMENTS

CLIP's training data is sourced from Internet, which means it rarely encounter data from certain specific domains. In order to verify Pelen's generalizability, we collect five datasets on three distinct image domain: medical, remote sensing and infrared images. To validate the effectiveness of the proposed prompt reverse learning (PeLen) method, we conduct two tasks using these five datasets.

### 4.1 SETUP AND BASELINES

We employ two highly representative recognition tasks and five datasets from three rare image domain to evaluate the effectiveness of different prompt learning methods.

**Image classification task.** This task aims to assign images to predefined categories. We divide the dataset into training, validation, and test set. Current prompt learning methods usually encounter challenges in rare image classification tasks because these images are rarely appear in the training set of CLIP. Therefore, the proposed PeLen method primarily focuses on image classification tasks.

**Domain adaptation task.** It is particularly important to adapt a pre-trained vision language model, e.g., CLIP, to a target domain with different characteristics, e.g., rare iamges. In the context of few-shot learning, the target domain has limited labeled data. We use 16 shots for training while test models on the full test set. Few-shot learning demonstrates the ability to transfer cross-domain knowledge, which is used to verify the effectiveness of PeLen.

**Datasets.** The above two tasks involve five datasets: ISIC2018 Codella et al. (2019); Tschandl et al. (2018), MURA Rajpurkar et al. (2017), RSSCN7 Zou et al. (2015), RGB-NIR Brown & Süsstrunk (2011) and a pediatric brain tumor dataset. ISIC2018 and MURA are medical image datasets used for melanoma detection and musculoskeletal abnormality detection respectively. RSSCN7 is remote sensing datasets, while RGB-NIR is infrared datasets respectively. Furthermore, we validated the effectiveness of PeLen in a private pediatric brain tumor dataset collected from hospital with adequate privacy protection measures.

**Baselines.** We compare our method with five existing prompt learning methods, including CoOp Zhou et al. (2022b) and Co-CoOp Zhou et al. (2022a), which add prompts on the text side, two visaul prompt method RLM-VP Bahng et al. (2022) and VPT Jia et al. (2022) and the latest multi-modal prompts method MaPLe Khattak et al. (2023).

## 4.2 IMPLEMENTATION DETAILS

**Metric.** Our experiment choose mean class accuracy as evaluation metric for ISIC2018 dataset, and accuracy for the other dataset. Due to the ISIC2018 dataset has severe long-tail distribution problem, the mean class accuracy considers the accuracy within each class, ensuring the final results is not biased by the varying number of samples across different classes.

**Data details.** The modality of medical images is different from natural images, so we need to process the data to be consistent with natural images. The images in MURA are single-channel X-rays, we copy the single channel three times. Besides, each patient has a different number of X-rays, we use a single sample when training the model, and vote on the results of multiple X-rays to get the final result for one patient during testing. Each patient in the pediatric brain tumor dataset has three modalities, we concatenate the three modalities, corresponding to the RGB channels of natural images. Additionally, since each modality has 24 frames of images, we intercept 8 consecutive frames of images starting from the third frame according to the location of the brain tumor to obtain the average feature vector for classification. The images in other datasets are consistent with natural images and do not require any special processing. The data augmentation strategies are employed during training including random horizontal flipping and random vertical flipping, randomly resize crop to $224 \times 224$, mean and standard deviation normalization.

**Experimental parameter details.** We perform PeLen on CLIP pre-trained ViT-B/16. In ISIC2018, RSSCN7 and pediatric brain tumor classification, the depth of the prompt is 12, the length is 10. In MURA, the depth of the prompt is 8, the length is 15. In RSSCN7, the depth of the prompt is 12, the length is 8. And the learning rate is 0.0005 in ISIC2018 and MURA dataset, 0.001 in other. All models are trained for 200 epochs with batch-size of 64 via adam Kingma & Ba (2014) optimizer on a single NVIDIA 3090 GPU. We report 5-fold cross-validation results on all tasks.

## 4.3 MAIN RESULTS

**Image classification experiments.** Table 1 shows the performance of PeLen on five rare image classification tasks, and PeLen achieves better performance on all five tasks. Although CoOp and Co-CoOp have powerful generalization capabilities for downstream tasks on natural images, their performance on rare image tasks is unsatisfactory. This also proves the huge gap between rare images and natural images, emphasizing the necessity of utilizing visual prompts to bridge this gap. Compared with RLM-VP and VPT, PeLen achieves huge performance gains. RLM-VP trains a box attached around the image as a visual prompt, and VPT uses an inserted token as a visual prompt. Although they both use visual prompts with additional label mapping in RLM-VP, PeLen allows the distribution of rare images downstream to be calibrated to a fixed upstream image distribution, demonstrating superior performance. PeLen only trains prompts in the image encoder. Compared with MaPLe, which uses multi-modal prompts, PeLen still improves the classification accuracy. In general, PeLen demonstrate excellent capabilities in multiple rare image domains and achieved a performance gain of 5.47% compared to the baseline on a real-world pediatric brain tumor dataset. Table 6 in Appendix shows the detailed results of different parts on the MURA dataset.

**Few-shot learning.** Table 2 shows the experimental results of PeLen and the state-of-the-art prompt learning methods under few-shot conditions. We use 16 shots for training and report results on the full test set. In these settings, PeLen demonstrates strong performance on three medical image

Table 1: **Comparison with state-of-the-art prompt learning methods.** We report the mean class accuracy (%) on the ISIC2018 dataset and accuracy on the other. On MURA dataset, We report average results across 7 body parts, detailed results are shown in Appendix. Our method has overall improvement. Bold numbers mean best and underlined are the second best.

| Method | ISIC2018 | MURA | RSSCN7 | RGB-NIR | Brain Tumor |
|---|---|---|---|---|---|
| CoOp | 61.68 | 76.44 | 91.64 | 84.68 | 50.71 |
| Co-CoOp | 56.91 | 75.09 | 94.25 | 86.17 | 37.26 |
| RLM-VP | 59.62 | 70.02 | 88.18 | 73.78 | 60.86 |
| VPT | 74.34 | 81.88 | 96.57 | 89.11 | 77.03 |
| MaPLe | 73.38 | 81.97 | 96.14 | 89.72 | 73.67 |
| PeLen (Ours) | **78.40** | **82.28** | **96.64** | **90.56** | **82.50** |

Table 2: **Prediction accuracy (%) under few-shot learning scenarios.** The table shows experimental results for 16 shots. Our method has strong robustness. Detailed results in MURA datasets are shown in Appendix. Bold numbers mean best and underlined are the second best.

| Method | 16 shots | | | | |
|---|---|---|---|---|---|
| | ISIC2018 | MURA | RSSCN7 | RGB-NIR | Brain Tumor |
| CoOp | 47.52 | 66.58 | 85.43 | 79.25 | 39.34 |
| Co-CoOp | 35.67 | 63.27 | 90.11 | 84.49 | 32.90 |
| RLM-VP | 32.13 | 58.11 | 70.14 | 67.07 | 45.87 |
| VPT | 55.21 | 68.14 | **92.37** | 87.06 | 56.67 |
| MaPLe | 51.22 | 68.56 | 91.29 | **87.31** | 52.62 |
| PeLen (Ours) | **55.76** | **70.50** | 92.11 | 86.81 | **60.51** |

datasets: ISIC2018, MURA and brain tumor datasets. Existing methods face challenges due to the data scarcity of medical images. However, the robustness observed across different medical image datasets highlights the potential applicability of PeLen in real-life clinical settings. Indeed, PeLen performs less favorably on the remote sensing image dataset RSSCN7 and the infrared image dataset RGB-NIR. This shows limitations in PeLen's application to these specific domains. Future research will delve deeper into understanding the underlying reasons for these differences and explore potential strategies for further improvement or modification.

## 4.4 Ablation Study

**Effectiveness of reverse learning.** We conduct ablation experiments for reverse learning on all five rare image tasks and Table 4 shows the detailed results. We take the baseline that only uses visual prompt with rare image class names as input of language branch, *i.e.*, VPT Jia et al. (2022). PeLen adds an additional reverse learning loss term based on the cross-entropy loss to calibrate the distribution of rare images, resulting in improving classification accuracy for all five tasks. However, using the reverse learning loss term alone does not yield performance gains. Since it only aligns the visual representation of rare images with common images but does not align the visual representation with the textual features of label-mapped class names.

**Impact of common class selection.** PeLen randomly selects common image classes and establishes mapping relationships with rare images. To assess the impact of common class selection on performance, we conducted experiments on the pediatric brain tumor dataset using two distinct groups of common image classes, each with completely different class names. The results in Table 3 demonstrate that random se-

Table 3: The experimental results of PeLen with two different random selected common class groups on pediatric brain tumor dataset.

| Common Class | ISIC2018 | Brain Tumor |
|---|---|---|
| Class Group 1 | 78.40 | 82.50 |
| Class Group 2 | 78.09 | 81.89 |
| Class Group 3 | 78.12 | 81.05 |

Table 4: **Ablation study.** We compare PeLen with the baseline, *i.e.*, VPTJia et al. (2022), on five tasks. VPT-RE uses only reverse learning loss, aligning rare images with common ones at the visual level. In contrast, PeLen maps and aligns rare and common images at both visual and textual levels. Results on the MURA dataset are averaged across 7 body parts.

| Method | $\mathcal{L}_{clip}$ | $\mathcal{L}_{re}$ | ISIC2018 | MURA | RSSCN7 | RGB-NIR | Brain Tumor |
|--------|------|------|----------|------|--------|---------|-------------|
| VPT    | ✓    |      | 74.34    | 81.88 | 96.57 | 89.11   | 77.03       |
| VPT-RE |      | ✓    | 68.88    | 80.84 | 94.00 | 80.73   | 76.22       |
| PeLen  | ✓    | ✓    | **78.40** | **82.28** | **96.64** | **90.56** | **82.50**   |

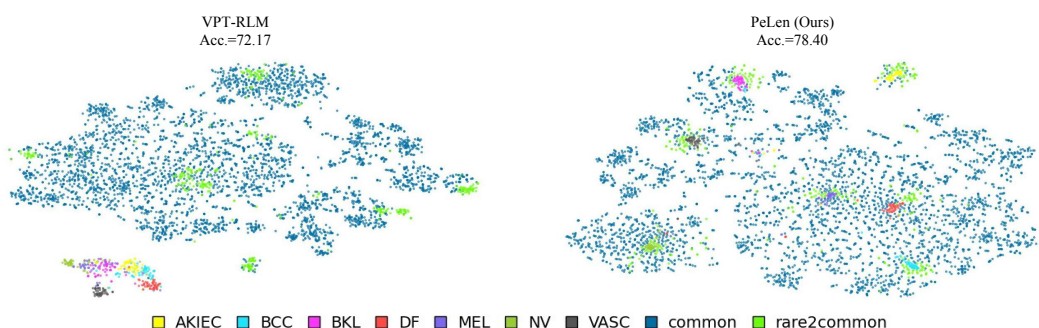

Figure 3: t-SNE visualization of visual representations of rare and natural images on ISIC2018 dataset. Blue dots stand for visual representations of natural images. Green dots represent the visual representation of natural images with the class name that corresponds to the class name of the rare image. Points of other colors represent visual representations of rare images. PeLen can integrate the representation of rare images into the representation space of natural images and increase the distinguishing ability.

lection of different common image classes does not affect performance. This further confirms the previous insight that CLIP is insensitive to the class names of rare images. Since CLIP lacks prior knowledge about rare images, PeLen calibrates rare image features into the common image representation space, where CLIP has strong perceptual capabilities.

## 4.5 VISUALIZATION

Figure 3 shows the visual representations space visualization of rare and common images on ISIC2018 dataset. VPT-RLM only uses the class names of common images to represent the class names of rare images, so the visual representation space of rare images is still independent of the representation space of natural images. However, CLIP lacks awareness of the rare image representation space. PeLen optimizes prompts to approach representations of rare images to those of common images, which aligns the visual representation of rare images with the visual and textual representations of natural images by category. This distribution calibration leads to full leverage of CLIP's powerful visual-textual matching capability.

## 5 CONCLUSION

Through prompt learning, large-scale vision-language models such as CLIP demonstrate strong generalization performance on downstream tasks with common images. But challenges arise when facing rare image tasks. We observe an intriguing phenomenon: CLIP lacks sensitivity to rare image class names. Leveraging this insight, we propose Prompt Reverse Learning. Differing from previous works, we align the representation of rare images with the fixed upstream representations of common images at both the visual and textual levels. To validate our approach's effectiveness, we conduct experiments on five datasets containing rare images. Our results demonstrate that Prompt Reverse Learning enhances the generalization ability of CLIP for rare image recognition tasks.

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

# A APPENDIX

## A.1 DATASET INTRODUCTION

**Melanoma detection.** Melanoma is a malignant tumor that accounts for 90% of skin cancer deaths Garbe et al. (2016). Current diagnostic methods for melanoma detection present several limitations and challenges. To address these problems, deep learning methods have emerged in melanoma detection in recent years. Here, we use the ISIC2018 datasets Codella et al. (2019); Tschandl et al. (2018). This dataset including 7 types of skin cancer. The training set has 10015 images, the validation set has 193 images, and the test set has 1512 images.

**Musculoskeletal abnormality detection.** Musculoskeletal abnormalities are the main cause of pain and disability Woolf & Pfleger (2003). Musculoskeletal abnormality detection is a binary task of whether X-ray images of the musculoskeletal system are normal. The MURA Rajpurkar et al. (2017) dataset contains a large number of X-ray images of multiple body parts, including elbow, finger, forearm, hand, humerus, shoulder, and wrist. The training set contains 8280 normal and 5177 abnormal, and the validation set contains 661 normal and 538 abnormal. Since the test set is not public, we test on the validation set.

**Remote sensing image classification.** This task has important applications in many fields, including agriculture, urban planning, environmental monitoring, resource management, etc. RSSCN7 Zou et al. (2015) dataset contains 2800 remote sensing images, which come from 7 typical scene categories: grass, field, industry, riverlake, forest, resident, parking.

**Infrared image classification.** Infrared image has significant applications in various fields such as military, security, meteorology, environmental monitoring, and medical imaging. For example, it can be used for night surveillance and border patrol to improve security and monitoring efficiency. RGB-NIR Brown & Süsstrunk (2011) dataset consists of 477 images in 9 categories captured in RGB and Near-infrared. The scene categories include: country, field, forest, indoor, mountain, oldbuilding, street, urban, water. We use infrared images in the dataset for training and testing.

**Pediatric brain tumor classification.** The incidence of pediatric brain tumors have jumped to the first place in the common childhood cancer Ostrom et al. (2020), seriously threatening children's lives and health. Pediatric brain tumors can be divided into different types according to their pathological properties, such as glioma and medulloblastoma. There are different clinical treatment options for different types of brain tumors. At present, the diagnosis and treatment of brain tumors are mainly based on pathological slices after surgery. This invasive detection method carries a high risk to patients as well as high financial and time costs. Therefore, it is of great significance to establish a non-invasive, convenient and accurate tumor classification method. We use the pediatric brain tumor dataset collected from the hospital. This dataset contains preoperative MRI images of 623 pediatric patients with posterior fossa tumors and the corresponding tumor types, including 201 medulloblastomas, 101 choroid plexus tumors, 177 ependymomas, and 144 gliomas. Each patient's preoperative MRI image had three modalities: T1-weighted, T1-weighted contrast-enhanced, and T2-weighted. Each modality consists of 24 frame images, each frame corresponding to a brain slice at a different location. The tumor is located in the posterior fossa, so only 8 frames are used during training. We preprocess the data as follows, applied the N4ITK Song et al. (2017) bias field correction method to remove the inhomogeneity of the bias field, and then applied HD-BET Schell et al. (2019) for brain extraction.

## A.2 CORRESPONDING TEXT PROMPT

Table 5 illustrates the description of the dataset and corresponding text prompts used in baseline method. In PeLen, all datasets use "a photo of a {}" as text prompt.

## A.3 EXPERIMENTAL RESULTS ON THE MURA DATASET

**Image classification experiment.** Table 6 shows the detailed results of the MURA dataset. PeLen achieves the best performance in five out of a total of seven parts compared to existing state-of-the-art prompt learning methods.

Table 5: Number of categories in the dataset and corresponding text prompt.

| Dataset | Classes | Text Prompt |
|---------|---------|-------------|
| ISIC2018 | 7 | "a photo of a {}" |
| MURA | 2 | "a photo of a {} X-ray" |
| RSSCN7 | 7 | "a photo of a {}" |
| RGB-NIR | 9 | "a photo of a {}" |
| Brain Tumor | 4 | "a photo of a {}" |

Table 6: Results for each body part on the MURA dataset are compared with state-of-the-art prompt learning methods. The last column shows the average results for the seven parts. Bold numbers mean best and underlined are the second best.

| Method | Body Parts in **MURA** | | | | | | | Average |
|--------|-------|--------|---------|------|---------|----------|-------|---------|
| | Elbow | Finger | Forearm | Hand | Humerus | Shoulder | Wrist | |
| Coop | 79.33 | 69.52 | 76.44 | 67.47 | 83.70 | 76.46 | 82.14 | 76.44 |
| Co-CoOp | 75.95 | 72.00 | 79.45 | 65.47 | 81.48 | 70.96 | 81.29 | 75.09 |
| RLM-VP | 73.42 | 72.00 | 64.66 | 66.27 | 69.14 | 70.10 | 74.54 | 70.02 |
| VPT | 84.18 | 78.86 | 81.20 | 78.84 | 83.95 | 80.07 | 86.08 | 81.88 |
| MaPLe | 84.18 | **80.57** | 80.45 | **79.24** | 83.95 | 79.04 | 86.38 | 81.97 |
| PeLen | **84.81** | 77.14 | 81.20 | 78.04 | **86.42** | **80.76** | **87.62** | **82.28** |

**Few-shot learning.** Table 7 shows the detailed experimental results of few-shot learning on the MURA dataset. Compared with state-of-the-art prompt learning methods, PeLen achieves the best performance in six out of a total of seven parts.

Table 7: Few-shot learning results for each body part on the MURA dataset. The last column shows the average results for the seven parts. Bold numbers mean best and underlined are the second best.

| Method | Body Parts in **MURA** (16 shots) | | | | | | | Average |
|--------|-------|--------|---------|------|---------|----------|-------|---------|
| | Elbow | Finger | Forearm | Hand | Humerus | Shoulder | Wrist | |
| Coop | 65.61 | 63.43 | 62.91 | 61.08 | 73.58 | 70.28 | 69.20 | 66.58 |
| Co-CoOp | 59.49 | 59.62 | 67.42 | 60.48 | 65.43 | 59.97 | 70.46 | 63.27 |
| RLM-VP | 55.49 | 61.14 | 56.64 | 59.28 | 59.75 | 54.12 | 60.34 | 58.11 |
| VPT | 66.46 | 67.24 | 73.18 | 62.68 | 70.62 | 64.95 | 71.87 | 68.14 |
| MaPLe | 66.88 | **69.52** | 74.44 | 60.68 | 71.61 | 66.50 | 70.32 | 68.56 |
| PeLen | **69.41** | 68.95 | **77.95** | **62.87** | **74.82** | **66.67** | **72.86** | **70.50** |

**Ablation study.** Table 8 shows the ablation study results on the MURA dataset. We choose VPTJia et al. (2022) as baseline. VPT-RLM adds label mapping based on the baseline, that is, specifying natural image class names to represent rare image class names. VPT-RE uses the reverse learning loss term alone based on VPT-RLM to align the visual representation of rare images with common images. PeLen also aligns the visual representation of rare images with the textual representation of label-mapped class names, yielding performance gains.

## A.4 EXPANDED VISUALIZATION

Figure 4 shows spatial visualization of visual representations of rare and natural images for two other datasets, the RSSCN7 and RGB-NIR dataset.

Table 8: The ablation study results on the MURA dataset. The last column shows the average results for the seven parts. Bold numbers mean best.

| Method | Body Parts in **MURA** | | | | | | | Average |
| | Elbow | Finger | Forearm | Hand | Humerus | Shoulder | Wrist | |
|---|---|---|---|---|---|---|---|---|
| VPT | 84.18 | **78.86** | **81.20** | **78.84** | 83.95 | 80.07 | 86.08 | 81.88 |
| VPT-RLM | 83.54 | 76.00 | 80.45 | 76.65 | 83.46 | 77.66 | 88.06 | 80.83 |
| VPT-RE | 82.91 | 77.91 | 76.70 | 78.04 | 86.67 | 78.01 | 85.65 | 80.84 |
| PeLen | **84.81** | 77.14 | **81.20** | 78.04 | **86.42** | **80.76** | **87.62** | **82.28** |

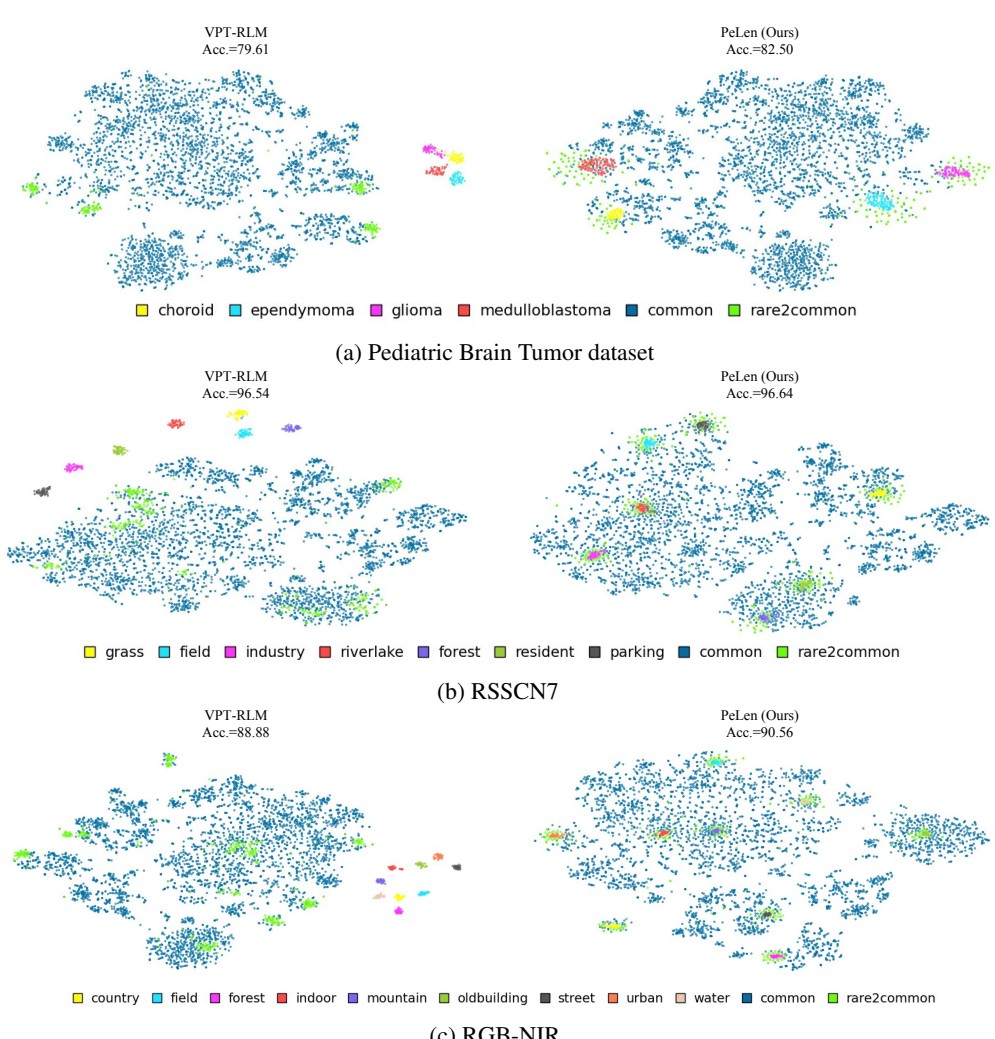

(a) Pediatric Brain Tumor dataset

(b) RSSCN7

(c) RGB-NIR

Figure 4: t-SNE visualization of visual representations of rare and natural images, from RSSCN7 and RGB-NIR dataset. Blue dots represent visual representations of natural images. Green dots represent the visual representation of natural images with class name that correspond to class name of the rare image. Points of other colors represent visual representations of rare images.

