# OpenReview forum: "Prompt Reverse Learning: Enhancing Visual Language Models for Rare Image Recognition"
_ICLR.cc/2025/Conference — ICLR 2025 Conference Withdrawn Submission_

### Official Review · Reviewer_muCR · 2024-10-28

**Soundness:** 3
**Presentation:** 3
**Contribution:** 3
**Rating:** 5
**Confidence:** 4

**Summary:**

To tackle the failure of vision language models like CLIP when faced with rare data, this paper proposes a novel prompt learning strategy, termed prompt reverse learning (PeLen). The main motivation of this paper is to transfer the representations of rate images into the representations of common images. Overall, this paper is well-written, but there are still some critical issues.

**Strengths:**

1. The writing of this paper is good and easy to follow.
2. The motivation of this paper is interesting.
3. The experimental results demonstrate its effectiveness.

**Weaknesses:**

1. Though the experimental results look good, the motivation of this paper is a little counterintuitive since the data distributions of rare images are obviously different from the data distributions of common images, the hard alignment of these two distributions seems to be unreasonable.

2. The mapping function between the rare class names and the common class names is not detailed described, which is very important for this paper. I suppose the correct mapping could seriously influence the performance.

3.  The compared methods are relatively old. The latest compared method is MaPLe, which was published in CVPR2023, maybe more recent works should be considered. Moreover, though the method claims to aim at rare image classification, it really looks like the Domain Prompt Learning concept, which is already published in [1,2]. Maybe these works should be added to consideration.

[1] Domain-controlled prompt learning

[2] Domain prompt learning with quaternion networks

**Questions:**

1. Could you give more explanations for the motivations, which could make the motivation more reasonable?

2. More details with regard to the mapping function should be given.

3. More related and recent prompt learning methods, especially the domain prompt learning methods, should be added for consideration.

If these issues are appropriately addressed, I would raise the rating scores.

---

> ### Author Response · Authors · 2024-11-22
>
> **Could you give more explanations for the motivations, which could make the motivation more reasonable?**
>
> Our task is to solve image recognition in a rare image domain with limited data. We observed that CLIP has a strong ability to represent images, though it is insensitive to class names for rare images. We propose a new paradigm to connect rare and common images, enabling CLIP to recognize rare images. In our method, the bridge is a mapping from the rare domain to the common domain, achieved by training a new image encoder. Our PeLen method not only maintains CLIP's simple pipeline but also improves recognition performance in the rare image domain
>
> **More details with regard to the mapping function should be given.**
>
> First, we randomly assign a common image class name to represent a rare image category, where we conduct multi-group assignment to validate its effectiveness. Second, we train a new model to align the rare image with the corresponding common image class as mapped above.

---

> > ### Comment · Reviewer_muCR · 2024-11-26
> >
> > Thanks for your response. After going through all the comments and responses, I tend to keep my score.

---

### Official Review · Reviewer_vxUN · 2024-10-30

**Soundness:** 3
**Presentation:** 3
**Contribution:** 2
**Rating:** 5
**Confidence:** 5

**Summary:**

This paper proposes an alternative prompt learning mechanism to improve the performance of CLIP in rare image categories. The paper finetunes a set of visual tokes and adapts task-specific representation with these tokens to CLIP’s representation. The paper outlines a clear method for this adaptation process, particularly focusing on rare image categories such as medical images and remote sensing data.

**Strengths:**

- The motivation is clear. Typically, prompt learning methods involve common categories while rare images from medical imaging or remote sensing possibly have complex label names which VLMs like CLIP failed to attenuate. The paper attempts to explore this problem.

- The paper is well written in a structured way with necessary improvement in figures.

**Weaknesses:**

1. The idea of leveraging CLIP's source representation for the target distribution of images is well explored in the literature [1,2,3]
The MR (1) learns an input perturbation program to mimic the source images (common in your case) and (2) does label mapping for the final prediction.  Particularly, [2] studies this adaption from a label mapping perspective.  How is the core idea of PeLen different from these works?  The visual tokens in PeLen replace the input perturbation program.

2. Please explain what is meant by 'Reverse Learning'. Does it mean you learn the prompts based on the image embeddings of the original CLIP?

3. Is label mapping essential for CLIP as it uses similarity-based selection? Could you provide details about its benefits for CLIP?

4. The t-SANE plots could be improved. It will be nice to see how the rare images utilize the decision boundaries associated with common categories at inference.

5. Including the CLIP Zero-shot performance on the tested datasets is better.

[1] Tsai, Yun-Yun, Pin-Yu Chen, and Tsung-Yi Ho. "Transfer learning without knowing: Reprogramming black-box machine learning models with scarce data and limited resources." International Conference on Machine Learning. PMLR, 2020.
[2] Chen, Aochuan, et al. "Understanding and improving visual prompting: A label-mapping perspective." Proceedings of the IEEE/CVF Conference on Computer Vision and Pattern Recognition. 2023.
[3] Bahng, Hyojin, et al. "Exploring visual prompts for adapting large-scale models." arXiv preprint arXiv:2203.17274 (2022).

**Questions:**

1. Explain how this is different from Model Reprogramming or Visual prompts for CLIP. The core idea looks the same except for the architectural modification.

2. Alternative mapping strategies such as frequency-based [1] can be used.

3. How does PeLen align rare and common images in both modalities? As mentioned in Table 4.

[1] Chen, Aochuan, et al. "Understanding and improving visual prompting: A label-mapping perspective." Proceedings of the IEEE/CVF Conference on Computer Vision and Pattern Recognition. 2023.

---

> ### Author Response · Authors · 2024-11-22
>
> **Explain how this is different from Model Reprogramming or Visual prompts for CLIP. The core idea looks the same except for the architectural modification.**
> Our approach differs from visual prompts, which add learnable prompts to the visual input. In contrast, our method transfers CLIP's understanding capabilities from common images to rare images through a mapping mechanism. Additionally, our method introduces no extra steps during inference compared to the original CLIP pipeline.
>
> **How does PeLen align rare and common images in both modalities? As mentioned in Table 4.**
> Our approach aligns the two modalities through a similarity loss function during training:
> $$L_{\text{re}}(\theta) = - \frac{1}{B} \sum_{i=1}^{B} \left( 1 - \text{sim}(x_{S_i}(\theta), x_{N_j}) \right).$$
> ] This loss function constrains the domain gap during training, aligning rare and common image features within the same space.

---

> > ### Comment · Reviewer_vxUN · 2024-11-26
> >
> > Thank you for your response.
> >
> > Here are my thoughts:
> > I believe BAR[1] performs a similar function to yours, as they utilize visual prompts instead of prefix tokens. Furthermore, their approach does not require an additional visual encoder.
> >
> > I have read your responses to the concerns raised by myself and other reviewers. I feel that the authors have only partially addressed these issues. While the paper presents an alternative proposal for domain adaptation and label mapping, it lacks critical evaluation. Therefore, I will maintain my current score.
> >
> > [1]Tsai, Yun-Yun, Pin-Yu Chen, and Tsung-Yi Ho. "Transfer learning without knowing: Reprogramming black-box machine learning models with scarce data and limited resources." International Conference on Machine Learning. PMLR, 2020.

---

### Official Review · Reviewer_L2YF · 2024-11-01

**Soundness:** 3
**Presentation:** 3
**Contribution:** 3
**Rating:** 5
**Confidence:** 4

**Summary:**

This paper explores the challenges of prompt learning for CLIP when applied to rare images, revealing that it is largely insensitive to the class-name initialization for such images. To address this, the authors propose a novel method called prompt reverse learning (PeLen), which adapts representations of rare classes by mapping them to common class representations. Specifically, PeLen uses common images and their class names to serve as proxies for rare images and names, effectively aligning their representations through prompt tuning and a reverse-loss mechanism. The experimental results demonstrate impressive gains for rare image recognition, particularly in medical domains.

**Strengths:**

1. The paper is well-written and easy to follow, with clear problem formulation and solution explanation.

2. Achieves strong performance on rare image datasets, showing promise for practical applications beyond the standard natural image domain.

3. Creative adaptation of CLIP to enhance rare image recognition, with the introduction of a novel prompt reverse learning strategy.

**Weaknesses:**

1. Lack of Baseline Comparisons: The study lacks comparisons with strong baseline approaches such as Adapter-based methods for CLIP (e.g., DCLIP[2], CLIP-Adapter[1], VDT-Adapter [3]). Including these baselines would help contextualize the benefits of PeLen relative to existing adaptation strategies.

2. Base-to-New Transfer Limitation: The proposed PeLen strategy does not seem straightforward for base-to-new transfer scenarios. Unlike CoOP and CoCoOp, which are capable of extending to unseen classes in the same domain, PeLen lacks an evident mechanism for handling novel classes without further prompt tuning. There is a need for clarity on how new classes or images could be mapped effectively into CLIP's embedding space without the need for labeled examples from the new domain.

3. Potential for Simplification (Ablation): The suggested method is akin to mapping the rare images and class names to well separated clusters in CLIP's joint embedding space. Could a simpler approach achieve similar results? Specifically, by using well-separated image prototypes of common concepts instead of the text embeddings in CLIP space, then use visual prompt tuning (VPT) with both the reverse loss and the cross entropy loss to align the rare images to these well separated prototypes. Here for cross entropy loss, rare image-to-common image image prototype similarity could be used as the logits.  If the text encoder is indeed insensitive to rare class names, then would this simpler approach suffice?



Missing Citations: Important related works, especially on Adapter-based approaches[1][3][4] and descriptive CLIP methods[2][3], are missing from the related works section. Adding these would improve the positioning of PeLen.

Minor Errors:

Line 189: It incorrectly suggests that CLIP uses Eqn 4 during training. CLIP employs a contrastive loss during pre-training, whereas the prompt tuning uses cross-entropy as outlined in Eqn 4. Clarification is needed here.

[1] Gao, Peng, et al. "Clip-adapter: Better vision-language models with feature adapters." International Journal of Computer Vision 132.2 (2024): 581-595.

[2] Menon, Sachit, and Carl Vondrick. "Visual classification via description from large language models." arXiv preprint arXiv:2210.07183 (2022).

[3] Maniparambil, Mayug, et al. "Enhancing clip with gpt-4: Harnessing visual descriptions as prompts." Proceedings of the IEEE/CVF International Conference on Computer Vision. 2023.

[4] Zhang, Renrui, et al. "Tip-adapter: Training-free clip-adapter for better vision-language modeling." arXiv preprint arXiv:2111.03930 (2021).

Overall I like the work due to its potential impact on practical domains, and I am willing to raise my score.

**Questions:**

1. Baseline Comparisons: Could you compare PeLen to Adapter-based baselines and descriptive text-based approaches? This would help establish its relative effectiveness and practicality.

2. Intuitions on Base-to-New Settings: Can you provide any intuition or experiments on how PeLen could potentially adapt to base-to-new class settings?

3. Ablation Studies: An ablation study where cross-entropy loss and reverse loss are applied just to the vision encoder—while using common image prototypes—would provide insight into the necessity of CLIP’s text encoder for aligning representations.

4. Dataset Release: Will the new dataset be made publicly available? This could foster future research and comparisons on rare image recognition tasks.

---

> ### Author Response · Authors · 2024-11-22
>
> **Intuitions on Base-to-New Settings: Can you provide any intuition or experiments on how PeLen could potentially adapt to base-to-new class settings?**
>
> PeLen randomly assigns a common image with label $y_k$ to the rare image with label $y_m$ to establish a mapping relationship between the images:
> $$
> H(I_{S_{i,k}}) = I_{N_{j,m}}, \quad I_{S_{i,k}} \in S, \, I_{N_{j,m}} \in N,
> $$
> where $I_{S_{i,k}}$ represents the image with the $k$-th class name of the rare image dataset, and $I_{N_{j,m}}$ represents the image with the $m$-th class name of the common image dataset. $H(\cdot)$ is the image mapping function.
>
> **Dataset Release: Will the new dataset be made publicly available? This could foster future research and comparisons on rare image recognition tasks.**
>
> We regret to inform you that due to the presence of sensitive privacy-related data, we are unable to make our private dataset publicly available. We appreciate your understanding in this matter.

---

> > ### Comment · Reviewer_L2YF · 2024-11-26
> >
> > Thank you your response. After going through the response I believe my questions have only been answered partially. Therefore I would like to maintain my score.

---

### Official Review · Reviewer_evHE · 2024-11-04

**Soundness:** 2
**Presentation:** 3
**Contribution:** 2
**Rating:** 5
**Confidence:** 3

**Summary:**

This paper presents a new approach to improve the performance of visual language models, such as CLIP, on tasks involving rare images. The authors identify that existing prompt learning methods struggle with rare data due to CLIP's insensitivity to class names of such images. To address this, they propose a new strategy called Prompt Reverse Learning (PeLen), which aligns representations of rare images with those of common images by leveraging a random mapping of common class labels to rare ones. Through experiments on various datasets, including medical, remote sensing, and infrared images, the authors demonstrate that PeLen enhances recognition performance for rare images compared to existing methods.

**Strengths:**

1.  The paper introduces a new approach called Prompt Reverse Learning (PeLen), which addresses the limitations of existing visual language models (like CLIP) in recognizing rare images. This approach creatively leverages the insensitivity of CLIP to rare class names by establishing random mappings between common and rare classes.
2. The experimental design is comprehensive, encompassing multiple datasets that cover diverse rare image categories, such as medical, remote sensing, and infrared images.
3.  The paper is well-organized, communicating complex ideas in an accessible manner. The structure facilitates understanding, with distinct sections outlining the motivation, methodology, and experimental results.
4. By enhancing the performance of models on rare images, the contributions of this paper could have impact on real-world applications, improving diagnostic tools and automated systems in domains where data scarcity is a challenge.

**Weaknesses:**

1. Lack clarity of Mapping Implementation and Rationale: The paper's discussion on the random mapping strategy between rare and common classes lacks clarity and depth in both its implementation and theoretical justification. While the authors propose this mapping to leverage CLIP's insensitivity to rare class names, the details of how this mapping works in practice are not entirely detailed. Furthermore, the paper does not provide a comprehensive analysis of how this approach fundamentally differs from existing visual prompt methods such as VPT(Jia et al. 2022) and RLM-VP(Bahng et al. 2022).

2. Limited Insights into Few-Shot Performance: Although the paper reports few-shot learning results, it lacks a thorough exploration of PeLen’s behavior across varying shot numbers (e.g., from 8 to 32). Such an analysis could explain how quickly PeLen adapts to new domains with limited data and whether it achieves stable performance improvements with fewer examples. Additionally, exploring few-shot performance under significant domain shifts (e.g., when target domains are visually dissimilar to the source) would reveal PeLen’s limitations and strengths, enhancing the understanding of its few-shot adaptation capabilities.

3. Lack of Parameter Sensitivity Analysis: PeLen’s effectiveness is notably impacted by hyperparameters, including prompt sentence example, but the paper does not provide an analysis of these parameters across datasets or settings. This makes it challenging for practitioners to replicate the results or apply PeLen to different datasets effectively.

4. Insufficient Evaluation on Domain-Specific Tasks: The paper evaluates PeLen exclusively on image classification tasks, which do not fully encompass the specific challenges posed by rare image domains like medical imaging, where tasks like segmentation and detection are also crucial. Rare image domains often require nuanced identification of small features within images (e.g., tumor boundaries in MRI scans), and segmentation or detection experiments could demonstrate PeLen's ability to handle these intricacies.

Reference

Hyojin Bahng, Ali Jahanian, Swami Sankaranarayanan, and Phillip Isola. Exploring visual prompts for adapting large-scale models. arXiv preprint arXiv:2203.17274, 2022.

Menglin Jia, Luming Tang, Bor-Chun Chen, Claire Cardie, Serge Belongie, Bharath Hariharan, and Ser-Nam Lim. Visual prompt tuning. In European Conference on Computer Vision, pp. 709–727.Springer, 2022.

**Questions:**

1. Justification of Random Mapping Strategy:  Can the authors elaborate on the rationale behind the use of a random mapping strategy for aligning rare and common classes?

2. Few-Shot Adaptation Capabilities: To better understand PeLen’s few-shot adaptability, have the authors investigated its performance across different shot settings (e.g., with 8, 16, or 32 shots)?

3. Comparisons with Specialized Few-Shot Learning and Adapter-Based Methods: Given PeLen’s similarity to methods like Tip-Adapter (Zhang et al., 2021), which employs an adapter module for few-shot adaptation without modifying model parameters, could the authors clarify how PeLen’s random mapping and prompt based alignment differ in effectiveness from adapter-based approaches?

4. CLIP Training Dataset Analysis and PeLen's Generalization Across Rare Domains: Has there been an investigation into the proportion of rare categories within CLIP’s original training dataset, such as in the WIT (WebImageText) dataset? Is it the case that the scarcity of these domains within the dataset leads to a reduced number of image features being learned, and is the few-shot learning approach employed to enhance the model's capacity to adapt?

5. Parameter Sensitivity and Recommendations: Since prompt depth and length impact PeLen’s performance, could the authors provide guidelines or recommended ranges for these parameters across various datasets?

6. Generalizability of PeLen framework: Have experiments been conducted across an even broader spectrum of rarer image domains (Astronomical Data, Satellite Imagery for Ecological Studies, Art Historical Artifacts etc.) to demonstrate the generalizability of the Prompt Reverse Learning (PeLen) framework, thereby validating its effectiveness and robustness in various real-world scenarios characterized by data scarcity?

---

> ### Author Response · Authors · 2024-11-22
>
> **Justification of Random Mapping Strategy: Can the authors elaborate on the rationale behind the use of a random mapping strategy for aligning rare and common classes?**
>
> Due to the scarcity of both text labels and images of rare examples in various pretrained model datasets, this mapping aims to transfer the perceptual abilities of pretrained models on common images to rare images. Our contribution focuses on a new paradigm to bridge the domain gap between rare and common images for recognition. The Random Mapping Strategy is an efficient and effective method for assigning labels.
>
> **Comparisons with Specialized Few-Shot Learning and Adapter-Based Methods: Given PeLen’s similarity to methods like Tip-Adapter (Zhang et al., 2021), which employs an adapter module for few-shot adaptation without modifying model parameters, could the authors clarify how PeLen’s random mapping and prompt based alignment differ in effectiveness from adapter-based approaches?**
>
> First, our PeLen method maintains the same number of weights as CLIP, making it more efficient than adapter-based methods. Second, we use a random mapping strategy for label assignment, introducing almost no additional overhead compared to the original CLIP.

---

### Note · Authors · 2024-12-02

I have read and agree with the venue's withdrawal policy on behalf of myself and my co-authors.